# Research on Cold Resistance of *Kandelia obovata* Transplanted to Zhoushan Area at the mRNA Level

**DOI:** 10.3390/ijms27010429

**Published:** 2025-12-31

**Authors:** Haozhe Li, Zhibin Sun, Weiye Li, Xiaolong Yin, Xian Xu, Xiaolin Zhang, Xiaojun Yan, Xinan Wang, Yuanyuan Li, Aijun Ma

**Affiliations:** 1Laboratory of Marine Biology Protein Engineering, Marine Science and Technical College, Zhejiang Ocean University, Zhoushan 316022, China; lihaozhe0601@163.com (H.L.); zhangxiaolin@zjou.edu.cn (X.Z.); yanxj@zjou.edu.cn (X.Y.); 2State Key Laboratory of Mariculture Biobreeding and Sustainable Goods, Qingdao Key Laboratory for Marine Fish Breeding and Biotechnology, Yellow Sea Fisheries Research Institute, Chinese Academy of Fishery Sciences, China-ASEAN Belt and Road Joint Laboratory on Mariculture Technology (Qingdao), Qingdao 266071, China; wangxa@ysfri.ac.cn; 3Laboratory for Marine Biology and Biotechnology, Qingdao Marine Science and Technology Center, Qingdao 266237, China; 4Zhoushan Fisheries Research Institute, Zhoushan 316000, China; liweiye2000@126.com (W.L.); xlyhndx@163.com (X.Y.); 17805806502@163.com (X.X.); 5CAS Center for Excellence in Molecular Plant Sciences, Chinese Academy of Sciences, Shanghai 200032, China; yyli@cemps.ac.cn

**Keywords:** *Kandelia obovata*, low-temperature stress, physiological mechanisms, transcriptome, SNP

## Abstract

To elucidate the physiological and molecular mechanisms underlying cold tolerance in the mangrove species *Kandelia obovata* Sheue & al, this study measured the antioxidant enzyme activities and photosynthetic pigment contents of two populations—cold-tolerant and -sensitive—under natural overwintering conditions. In addition, transcriptome sequencing was performed to analyze differentially expressed genes (DEGs), transcription factor families, single nucleotide polymorphisms (SNPs), and alternative splicing events. The results showed that catalase activity was significantly elevated in the cold-tolerant population, which enhanced the efficiency of hydrogen peroxide scavenging. In contrast, although the superoxide dismutase activity was relatively high in the cold-sensitive population, its downstream scavenging capacity was insufficient, resulting in an overall lower antioxidant efficiency. The KEGG enrichment analysis indicated that pathways such as phenylpropanoid biosynthesis, amino sugar metabolism, and plant hormone signal transduction might be involved in the response to low-temperature stress. Further analysis revealed that transcription factors such as WRKY, NAC, MYB, and ERF were differentially expressed at significant levels in the cold-tolerant population, suggesting that they may play important roles in low-temperature adaptation. In addition, the diversity of SNPs and alternative splicing events may enhance protein function and contribute to improved cold tolerance. In summary, the cold-tolerant *K. obovata* population achieves low-temperature tolerance through multiple mechanisms, including antioxidant defense, metabolic regulation, and transcriptional as well as post-transcriptional regulation. This study provides a theoretical basis for elucidating the molecular foundations of cold tolerance in *K. obovata*.

## 1. Introduction

Mangroves, tropical or subtropical intertidal plants, are primarily distributed along coastal shallows or estuarine mudflats at the interface between land and sea, forming evergreen woody plant communities dominated by mangrove species [1]. In China, mangroves are mainly found along the southern coastal provinces, including Hainan, Guangdong, Guangxi, Fujian, and Zhejiang [2], exhibiting characteristics of both terrestrial and marine plant communities. Mangrove species are light-demanding [3], salt- and alkali-tolerant [4,5], but sensitive to low temperatures [6]. They can purify water [7], filter organic matter and pollutants, promote sediment deposition [8], and play important roles in coastal protection as well as in resisting tides and wave action [9]. Mangroves also play a critical role in ecosystems, earning the reputation of “coastal guardians and green lungs of the ocean.” They provide important habitats for rare and endangered waterfowl, as well as breeding grounds for fish, shrimp, crabs, and shellfish [10], offering both significant economic value and ecological benefits.

*Kandelia obovata*, of the family Rhizophoraceae, is the most widely distributed and cold-tolerant mangrove species in China, occurring at the highest latitudes among mangroves [11]. However, its distribution is still limited by low temperatures. It is excellent material for studies on high-latitude introduction and cultivation. However, due to global warming and anthropogenic disturbances [12], natural mangroves are facing the risk of decline or even disappearance [13]. Temperature can influence the distribution of temperature-sensitive plants such as *K. obovata* [14]. Extreme low-temperature events during winter in Zhejiang can cause frost damage to *K. obovata*, affecting its overwintering ability. Historical records indicate that multiple extreme cold wave events in southern China in 2008 had severe negative impacts on local mangroves [11]. Even more seriously, in 2010, an extremely low-temperature cold wave in the Ximen Island area of Wenzhou caused severe frost damage to introduced *K. obovata* plants, resulting in inhibited photosynthetic pigment synthesis, disrupted antioxidant systems, and ionic imbalances in trees of different ages [15]. Therefore, to achieve successful introduction of *K. obovata* in northern Zhejiang, it is particularly urgent and critical to investigate its cold tolerance.

Previous studies have examined the effects of low temperatures on photosynthesis and antioxidant enzyme activities in mangrove species such as *Bruguiera gymnorrhiza*, *K. obovata*, and *Rhizophora stylosa* [16]. The results indicate that severe low-temperature stress can significantly reduce leaf net photosynthetic rates and stomatal conductance in mangroves. Meanwhile, the activities of enzymes superoxide dismutase (SOD) and peroxidase (POD) decrease significantly, and malondialdehyde (MDA) content increases markedly. Under low-temperature stress, chlorophyll content in mangrove leaves also declines, thereby affecting normal photosynthetic processes. Current studies on *K. obovata* have focused on seedling propagation [17], carbon sequestration [18], and physiological mechanisms, with limited investigations into the molecular mechanisms underlying cold tolerance. High-throughput sequencing combined with physiological analyses can provide a better understanding of the cold tolerance mechanisms in *K. obovata*.

In this study, researchers conducted long-term observations of *K. obovata* seedlings in Zhoushan, Zhejiang Province, including propagation and introduction. Based on data analysis, post-winter individuals were broadly classified into two groups: those that survived the winter, designated as the cold-tolerant group, and those that died after winter, designated as the cold-sensitive group. Although both groups originated from the same batch of *K. obovata* seedlings, they may derive from different mother plants and possess genetic differences, which could lead to distinct cold tolerance during winter [19]. In the cold-tolerant group, leaves remained healthy and gradually recovered after winter, allowing the plants to survive; in contrast, individuals in the cold-sensitive group either withered and died directly or exhibited slight leaf yellowing and failed to recover from cold-induced damage, ultimately leading to death. In this study, leaves from both cold-tolerant and -sensitive *K. obovata* groups were selected for physiological measurements and transcriptome sequencing. By assessing differences in enzyme activities and photosynthetic pigment contents between individuals with distinct cold tolerance, we aimed to investigate the functional roles of enzymes critical in the cold response of *K. obovata*. At the same time, key differentially expressed genes (DEGs) were identified, and the metabolic pathways and transcription factor (TF) families they regulate were explored to uncover potential genetic differences. This study combines physiological assessments and transcriptome profiling of *K. obovata* under natural overwintering conditions. Comparative analyses of antioxidant enzyme activities and mRNA-level gene expression between cold-tolerant and -sensitive plants provide preliminary insights into the molecular mechanisms underlying cold resistance. Furthermore, the exploration of potential SNPs and AS events may provide a theoretical foundation for the introduction of *K. obovata* to higher-latitude regions and for future selective breeding efforts.

## 2. Results

### 2.1. Comparison of the Survival Rate of Kandelia obovata After Overwintering

During the entire overwintering period, survival and mortality rates of *K. obovata* in Groups A and B were recorded and analyzed (Table 1). The cold-tolerant subgroup A1 had 1339 surviving individuals, comparable to the 1351 survivors in group B. The total number of deaths in group B was 1649, which was 74 higher than the total deaths in group A; these individuals likely died immediately during the cold wave and thus did not possess cold tolerance. Comparison of the survival rates between subgroup A1 and the entire group B demonstrates the feasibility of manually classifying *K. obovata* seedlings into cold-tolerant and -sensitive groups during the overwintering period.

### 2.2. Physiological Differences Between K. obovata Populations Under Low-Temperature Stress After Overwintering

To verify the physiological differences between the cold-tolerant and -sensitive *K. obovata* populations in response to low-temperature stress, the photosynthetic pigments and antioxidant systems in their leaves were analyzed. No significant differences were observed in chlorophyll contents between the two groups (Table 2). The levels of chlorophyll a, chlorophyll b, and total chlorophyll were statistically comparable (*p* > 0.05), suggesting that photosynthetic pigments were not the main factors contributing to the difference in cold tolerance between the two populations during the early overwintering stage. However, significant differences were observed between the antioxidant systems of the two groups. The CAT activity in the cold-tolerant group was 14.72 ± 0.19 U/g, significantly higher than in the cold-sensitive group (*p* < 0.05). As a key enzyme responsible for removal of hydrogen peroxide (H_2_O_2_), the higher activity of CAT indicates that the cold-tolerant population possesses a stronger reactive oxygen species (ROS) scavenging capacity under low-temperature stress, thereby reducing lipid peroxidation damage to cell membranes. In contrast, the SOD activity in the cold-sensitive group was 1435.10 ± 23.89 U/g, significantly higher than in the cold-tolerant group (*p* < 0.05), indicating a stronger capacity for scavenging superoxide anions (O_2_^−^). This difference may be associated with an immature cold-resistance mechanism in the cold-sensitive population, which tends to rely more on the early-stage removal of ROS but lacks an efficient pathway for subsequent of H_2_O_2_ decomposition. The POD activity was non-significantly lower in the cold-tolerant than the cold-sensitive group. Similarly, the MDA content was higher in the cold-sensitive group but showed no significant difference between the two populations. These results suggest that the degree of lipid peroxidation and certain antioxidant pathways did not differ substantially between the groups.

### 2.3. Quality Control Analysis of Raw Sequencing Data

Transcriptome sequencing was performed on a total of six samples, including both the experimental and control groups, generating 42.02 Gb of raw data. After removing low-quality reads and adapter sequences, the range of clean reads per sample was 42,780,366–50,516,336 (Table 3), with an error rate of approximately 0.01%. The Q20 values all exceeded 98.9%, Q30 values exceeded 96.45%, and the GC content was 45.45–45.93%. The clean reads of each sample were aligned to the *K. obovata* reference genome (version: GWHACBH00000000.1; source: https://ngdc.cncb.ac.cn/gwh/Assembly/20708/show (accessed on 4 December 2025)), and the mapping results are summarized in Table 4. The range of the total mapping rate was 97.79–98.12%, of the multiple-mapped rate was 3.16–3.96%, and of the uniquely mapped rate was 93.83–94.97%. These results indicate that the transcriptome sequencing data are of high quality and reliable for subsequent analyses.

### 2.4. Differential Gene Expression Analysis

A total of 19,197 expressed genes were detected: 17,264 known and 1933 novel genes. Principal component analysis (PCA) of the six samples from the cold-tolerant and -sensitive groups (Figure 1) clustered the three samples from the cold-tolerant group together, whereas the cold-sensitive samples displayed a more dispersed pattern. This may be because all samples originated from the same batch of naturally overwintered plants, and leaf selection did not specifically target visibly green or nearly senescent leaves. Since the cold stress response is not equivalent to senescence, PCA cannot perfectly separate the two groups. Nevertheless, this does not compromise the reliability of the results.

Based on the quantified gene expression levels, DEG analysis between the two groups was performed using DESeq2. Genes with |log2 fold change (FC)| ≥ 1 and *p*-value < 0.05 were considered to be DEGs. The volcano plot (Figure 2) showed a total of 246 identified DEGs, including 224 significantly downregulated and 22 significantly upregulated genes.

### 2.5. GO and KEGG Annotation and Enrichment Analysis of Differentially Expressed Genes

The purpose of Gene Ontology (GO) annotation is to classify the functions of target genes or transcripts. Using the GO database, genes and their products are functionally categorized. Results of enrichment and annotation analysis of the DEGs are shown in Figure 3, with the y-axis representing the second-level GO terms, the x-axis indicating the number of genes mapped to each term, and the colors denoting the three main GO categories. At the second-level classification, the DEGs were mainly annotated to cellular anatomical entity, catalytic activity, cellular process, binding, and metabolic process (Figure 3A). At the third-level classification, DEGs were predominantly associated with intrinsic component of membrane, organic substance metabolic process, and ion binding (Figure 3B). At the fourth-level classification, DEGs were mainly annotated to integral component of membrane, cation binding, and intracellular organelle (Figure 3C).

The GO enrichment analysis of the DEGs was performed using GOATOOLS, and significance was assessed with Fisher’s exact test. The GO terms with Padjust < 0.05 were considered significantly enriched. In Figure 3D, the y-axis represents the GO terms, while the x-axis indicates the Rich Factor, defined as the ratio of the number of DEGs mapped to a GO term to the total number of annotated genes. A higher Rich Factor indicates greater enrichment. The size of the bubbles corresponds to the number of DEGs in each GO term, and the color represents different Padjust ranges. The most significantly enriched terms included xyloglucan metabolic process, apoplast, and iron ion binding, while terms with the largest number of DEGs include oxidoreductase activity and extracellular region (Figure 3D).

Using the KEGG database, DEGs were classified according to the pathways they participate in or the functions they perform. The DEGs were mainly annotated to pathways related to metabolism and plant hormone signal transduction (Figure 4 and Table 5). In terms of gene number, the highest enrichment was in phenylpropanoid biosynthesis, followed by plant hormone signal transduction, and then amino sugar and nucleotide sugar metabolism. Regarding significance, DEGs in *K. obovata* leaves under low-temperature stress were most significantly enriched in phenylpropanoid biosynthesis, followed by amino sugar and nucleotide sugar metabolism, with plant hormone signal transduction ranking third.

### 2.6. Transcription Factor Prediction and qRT-PCR Validation

A total of 38,624 transcripts were detected in the transcriptome sequencing: 16,018 known and 22,606 novel transcripts. Prediction of TFs, using the Plant Transcription Factor Database (PlantTFDB), revealed that 2519 genes encode putative TFs, which were annotated into 1314 TFs belonging to 47 TF families (Figure 5). The 10 TF families with the largest number of genes were MYB-related, MYB, bHLH, ERF, HB-other, NAC, WRKY, bZIP, B3, and GRAS. Among these, the first eight families have been reported to play important roles in plant cold tolerance.

Based on the DEGs identified in the volcano plot (Figure 2), a total of 246 genes were differentially expressed between the cold-tolerant and -sensitive groups. Among them, 20 DEGs belong to 10 different TF families (Figure 6), with MYB, WRKY, ERF, NAC, and MYB-related previously demonstrated to be associated with cold tolerance.

To validate the reliability of the transcriptome data, seven DEGs (*geneMaker00014971*, *geneMaker00007720*, *geneMaker00006849*, *geneMaker00008016*, *geneMaker00000397*, *geneMaker00004010*, and *geneMaker00008573)* were randomly selected for qRT-PCR verification: three upregulated and four downregulated genes. The relative expression patterns of these genes were consistent with the transcriptome sequencing results (Figure 7), indicating that the transcriptome data generated in this study are accurate and reliable.

### 2.7. Alternative Splicing and SNP Analysis

From the transcriptome sequencing results of the cold-tolerant and -sensitive groups, five types of differential alternative splicing (AS) events were identified (Figure 8). Among them, skipped exon (SE) was the most prevalent, with a total of 385 events, accounting for 50.79%, followed by alternative 5′ splice site (A5SS, 17.15%), alternative 3′ splice site (A3SS, 17.02%), and retained intron (RI, 10.42%), while mutually exclusive exons (MXE) were the least frequent, representing only 4.62%. The SE was the most significant AS pattern in terms of differential expression, comprising 172 significant exclusion events and 213 significant inclusion events (Figure 9). The A5SS and A3SS events also exhibited inclusion/exclusion differences between the two groups, whereas the numbers of RI and MXE events were comparatively lower.

Single nucleotide polymorphisms (SNPs) refer to variations at a single nucleotide position in a DNA sequence, including transitions and transversions. In this study, transcriptome sequencing was performed on three samples each from the cold-tolerant and -sensitive *K. obovata* groups to analyze gene expression differences. The sequencing results were aligned to the reference genome, and are presented in Figure 10 and Table 6. The SNP type distribution patterns were largely consistent between the cold-tolerant and -sensitive groups. A total of 11,495–12,982 SNP sites were identified. The number of transition mutations (Ti; A/G, C/T, G/A, and T/C) was significantly higher than that of transversion mutations (Tv; A/C, G/C, C/A, A/T, C/G, G/T, T/A, and T/G), with Ti/Tv ratios ranging within 1.24–1.30. Further comparison showed that C/T and G/A transitions were the most abundant, with each sample containing over 1800–2000 sites, whereas transversions were less frequent, with range 428–971. Overall, the total number of SNPs was slightly higher in the cold-tolerant than the cold-sensitive group.

By comparing the SNP sites between the cold-tolerant and -sensitive groups, a total of 35 genes with significant SNP differences were identified. Integrating the transcriptome expression data revealed significant differences in KEGG metabolic pathways for *geneMaker00005577* and *geneMaker00013925*. The former is primarily involved in Phenylpropanoid biosynthesis, flavonoid biosynthesis, and stilbenoid, diarylheptanoid and gingerol biosynthesis, catalyzing the conversion of caffeoyl-CoA to feruloyl-CoA, representing a key node for lignin, flavonoid, and gingerol synthesis. This gene harbors a G → A mutation in the 5′ UTR and is significantly downregulated. The latter is mainly involved in the Monoterpenoid biosynthesis pathway, catalyzing the reversible conversion between menthone and neomenthol. This gene carries a nonsynonymous mutation in the exon region, causing the amino acid property to change from hydrophobic to polar, and is also downregulated. The mutations and altered expression of these two genes may reduce the flux of defense-related metabolic pathways, thereby affect the synthesis of antioxidants and signal molecules and regulate the physiological and metabolic adaptation of *K. obovata* under low-temperature stress.

## 3. Discussion

### 3.1. Physiological Differences Between Groups

When plants are exposed to low-temperature conditions, membrane lipid fluidity and metabolic rates decrease, leading to impaired electron transport and excessive production of ROS, including O_2_^−^, H_2_O_2_, and hydroxyl radical (‧OH). The accumulation of ROS can damage membrane lipids, proteins, and pigments, resulting in the formation of the lipid peroxidation product MDA. Simultaneously, the photosynthetic system is compromised, chloroplast membrane structures are disrupted, and chlorophyll degradation occurs. Under low-temperature stress, mangrove plants maintain photosynthesis through photosynthetic pigments [20] and simultaneously rely on the antioxidant enzyme system to scavenge ROS, thereby coping with environmental stress [21]. Chlorophyll a and chlorophyll b are the primary pigments in photosynthesis, and are directly involved in light absorption and electron transport, while total chlorophyll content provides an overall measure of the photosynthetic pigment level.

The results of this study showed no significant differences between the two groups of *K. obovata* in chlorophyll a, chlorophyll b, or total chlorophyll content (*p* > 0.05). This is mainly related to the sampling criteria of the experiment: the cold-tolerant and -sensitive groups exhibited only minor differences in leaf status at the early stage. After overwintering, as ambient temperatures gradually increased, the cold-tolerant *K. obovata* plants progressively resumed growth, whereas the cold-sensitive group failed to recover due to severe damage. Consequently, the physiological differences between the two groups increased over time. This divergence may not only result from differential responses to low temperature stress, but also reflect differences in survival status. Therefore, sampling was conducted during the early recovery stage after overwintering in this study.

Under low-temperature stress, plants rapidly activate the antioxidant defense system to scavenge ROS through various enzymatic reactions [22] (Figure 11). The first antioxidant enzyme to act within the ROS scavenging system is SOD, catalyzing the dismutation of O_2_^−^ into H_2_O_2_ thereby reducing oxidative damage to cells [23]. Enzyme CAT is key in removing H_2_O_2_ and preventing its accumulation from damaging cell membranes. Previous studies have shown that CAT activity in mangroves increases under cold conditions [14], which facilitates H_2_O_2_ scavenging and helps maintain membrane stability. Another important enzyme in the plant antioxidant system is POD, capable of eliminating excess ROS, and plays a crucial role in maintaining normal physiological functions and enhancing tolerance to low-temperature stress [21]. The final product of membrane lipid peroxidation, MDA, reflects the extent of cellular membrane damage and serves as an important indicator of plant cold tolerance [24]. The accumulation of MDA in plants is closely correlated with their cold tolerance: higher MDA levels indicate more severe oxidative damage to cell membranes and, consequently, lower cold tolerance [25].

The results of this study indicate that during the initial phase of antioxidant defense, SOD activity was significantly higher in the cold-sensitive than the cold-tolerant group, suggesting that the cold-sensitive population primarily relies on early-stage O_2_^−^ scavenging under low-temperature conditions, while downstream H_2_O_2_ decomposition may be insufficient, limiting overall antioxidant efficiency. These findings are consistent with previous studies [26]. During the adaptation phase to low-temperature stress, CAT activity was significantly higher in the cold-tolerant than the cold-sensitive group, indicating that the cold-tolerant group can decompose H_2_O_2_ more efficiently, thereby enhancing antioxidant defense under low temperatures. In contrast, POD activity did not differ significantly between the two groups (*p* > 0.05), possibly because this enzyme plays a limited role in the low-temperature response or functions more in a synergistic manner. Although MDA content did not differ significantly between groups, the cold-tolerant group exhibited lower MDA levels, indirectly reflecting stronger leaf antioxidant capacity and higher cold tolerance, consistent with the above results.

In summary, cold-tolerant *K. obovata* populations may enhance overall antioxidant defense under low temperatures primarily by increasing CAT activity and thereby improving H_2_O_2_ scavenging capacity. In contrast, although SOD activity is higher in cold-sensitive populations, their downstream H_2_O_2_ decomposition may be insufficient, resulting in lower overall antioxidant efficiency and contributing to interpopulation differences in cold tolerance. Differences in POD activity and MDA content were not significant, possibly because certain antioxidant pathways and levels of lipid peroxidation contribute only minimally to the observed variation in cold tolerance between populations. Another possible reason for this non-significant difference was the relatively limited sample size (*n* = 3) inherent to the experimental design.

### 3.2. Differential Expression Between Groups

The DEGs in *K. obovata* leaves under low-temperature stress exhibited distinct distribution patterns across various levels of GO annotation (Figure 3A–C). The GO enrichment analysis (Padjust < 0.05) revealed that the most significantly enriched functions were xyloglucan metabolic process, apoplast, and iron ion binding, while the functions with the highest number of DEGs were oxidoreductase activity and extracellular region (Figure 3D). These findings suggest that *K. obovata* may maintain cellular structure by regulating cell wall polysaccharide metabolism [27], modulate stress signaling through extracellular region-associated genes, and regulate ROS and ion homeostasis via oxidoreductase and metal ion-binding genes.

The KEGG annotation results indicated that DEGs were primarily enriched in metabolic pathways and hormone signal transduction-related pathways (Figure 4A). Significance analysis revealed that the phenylpropanoid biosynthesis pathway was the most significantly enriched, followed by amino sugar and nucleotide sugar metabolism and plant hormone signal transduction (Figure 4B). Previous studies have shown that pathways such as phenylpropanoid and jasmonic acid biosynthesis [28] are more readily activated under severe stress. During stress, the activation of starch and sucrose metabolism, the TCA cycle, glycolysis, and the pentose phosphate pathway can provide carbon sources and ATP for the synthesis of amino acids such as phenylalanine, valine, and threonine, thereby maintaining normal metabolic processes [25], which is consistent with our findings.

The GO and KEGG analyses indicate that under low-temperature stress, *K. obovata* leaves employ a multilayered response mechanism involving membrane protection, regulation of cell wall metabolism, ROS scavenging, maintenance of ion homeostasis, and hormone signal transduction. The significant enrichment of xyloglucan and amino sugar metabolic pathways suggests that *K. obovata* may enhance membrane and wall stability by regulating cell wall polysaccharides, thereby maintaining cellular structure under low-temperature stress. Genes related to iron and cation binding indicate that ion homeostasis plays a role in signal transduction and defense mechanisms. Furthermore, the enrichment of phenylpropanoid biosynthesis and plant hormone signaling pathways implies that *K. obovata* may coordinate the expression of defense genes through secondary metabolite production and hormone regulation, enabling metabolic adjustment and cold tolerance under low-temperature conditions.

Overall, the molecular response of *K. obovata* under low-temperature stress exhibits a multilayered, coordinated regulation, in which structural maintenance, redox regulation, and signaling pathways act synergistically to enhance cold tolerance. These findings provide a molecular basis for a deeper understanding of the low-temperature adaptation mechanisms in *K. obovata*.

### 3.3. Differential Transcription Factor Expression Between Groups

Gene expression regulation has a central role for TFs, and they participate in various physiological and biochemical processes and modulate plant responses to environmental stresses. In-depth investigation of TFs allows a more comprehensive understanding of the physiological responses and adaptive mechanisms employed by individual plants and their populations in response to environmental changes. Through transcriptome sequencing of annual *K. obovata* leaves, this study identified significant differences in gene expression between cold-tolerant and -sensitive groups. Such TFs as WRKY, NAC, MYB, bHLH, and ERF exhibited marked differential expression between the two groups, consistent with their established roles in plant cold tolerance [29]. Previous studies have shown that the TF WRKY40 can enhance cold tolerance in transgenic *Arabidopsis thaliana* [26]. The NAC family TFs have also been confirmed to participate in regulating plant low-temperature tolerance. For example, the *AmNAC11* gene from *Ammopiptanthus mongolicus* is induced by drought, cold, and high-salinity stress, and its expression in *Arabidopsis thaliana* via Agrobacterium-mediated transformation significantly improves tolerance to freezing and drought stress [30]. The bHLH TFs make a significant contribution to *K. obovata* cold tolerance, and the expression of *kobHLH* genes is associated with responses to cold stress [31]. Additionally, MYB [32] and ERF [33] TFs similarly promote physiological adaptation to low-temperature conditions by regulating the expression of cold-responsive genes.

The results of this study indicate that the high expression of certain TFs in the cold-tolerant *K. obovata* may activate a series of downstream defense responses, such as enhanced antioxidant enzyme activity and accumulation of osmolytes, thereby improving plant adaptation to cold environments (Figure 11). In the cold-sensitive group, reduced expression of these TFs may limit downstream defense responses, rendering the plants more susceptible to low-temperature stress. In summary, our transcriptome analysis and qRT-PCR validation demonstrates that differences in cold tolerance in *K. obovata* are closely associated with the expression of specific TF families, providing important insights into the molecular mechanisms underlying cold resistance.

### 3.4. Genetic Variation Between Groups

Both SNPs and AS play key roles in plant responses to abiotic stress, serving as important mechanisms for environmental adaptation and regulation of gene expression. Previous genome-wide association studies in rice have identified genes involved in responses to low-temperature stress and chilling accumulation, and screened for significantly associated SNPs [34]. In this study, SNPs at the transcriptome level were analyzed, revealing that cold-tolerant and sensitive *K. obovata* groups exhibited largely similar mutation types. Mutations in coding regions and transcriptionally active regions were predominantly transitions, with a Ti/Tv ratio stable around 1.3. This feature is consistent with SNP patterns reported in most plant transcriptomes or genomes. Further analysis indicated that C/T and G/A transitions were dominant, suggesting involvement of processes such as cytosine methylation deamination and RNA editing [35]. As a mangrove species evolved in coastal environments, *K. obovata* is frequently exposed to high salinity, hypoxia, and dramatic temperature fluctuations, which may promote the accumulation of C/T and G/A mutations through DNA methylation [36] and post-transcriptional regulatory mechanisms.

Combined analysis of SNP variation and transcriptome expression revealed that two genes in the cold-tolerant and -sensitive *K. obovata* groups harbored a 5′ UTR mutation and a nonsynonymous mutation in the exon region, respectively, and exhibited significant enrichment in KEGG metabolic pathways (Figure 11). A 5′ UTR mutation may weaken TF binding or alter mRNA secondary structure stability, leading to gene downregulation. This change could reduce lignin and flavonoid biosynthesis, thereby decreasing the accumulation of antioxidants and structural defense metabolites, ultimately affecting cold tolerance. Previous studies have also shown that the biosynthesis of phenylpropanoids and flavonoids contributes to enhancing cold tolerance in mangrove species [37], consistent with our findings. Alternatively, it may represent a regulatory strategy of “energy redistribution,” whereby structural metabolism is temporarily suppressed under low-temperature conditions to maintain a stable energy supply for critical cold-responsive metabolic processes. The nonsynonymous mutation in the exon region alters the amino acid property (hydrophobic → polar), which is predicted to affect the enzyme’s substrate-binding site or stability, thereby influencing catalytic efficiency and reducing the biosynthesis rate of monoterpenes. For example, the production of monoterpene defense compounds, such as volatile monoterpenes, may be suppressed, subsequently weakening signal transduction and the intensity of defense responses. Taken together, the combined analysis of SNP variation and transcriptome expression suggests that mutations in *geneMaker00005577* and *geneMaker00013925* may influence physiological metabolism and cold tolerance in *K. obovata* under low-temperature stress by modulating defense-related metabolic fluxes, the antioxidant system, and signaling processes.

Statistical analysis of the number and types of AS events between the two groups revealed substantial differences in AS patterns between cold-tolerant and sensitive *K. obovata* populations, with SE events being predominant (Figure 11), consistent with previous studies. Exon skipping can directly alter coding region length or cause loss of protein domains, representing one of the most functionally impactful splicing types, suggesting a potential key role in the formation of cold tolerance in *K. obovata*. Further integration of the results from Figure 8 and Figure 9 indicated that SE, A5SS, and A3SS events exhibited significant differences in both inclusion and exclusion counts, implying not only quantitative differences in AS events between the two groups but also a trend toward pattern conversion. These findings suggest that certain genes in the cold-tolerant population may preferentially undergo SE, and such differential splicing patterns could serve as molecular markers for cold tolerance.

Taken together, AS events in the *K. obovata* transcriptome provide an additional layer of regulatory mechanisms for low-temperature adaptation during its northward migration. Consistent with the SNP results, transcript-level variation not only manifests as nucleotide substitutions but also affects protein structure and function through altered splicing patterns, offering important insights into the molecular mechanisms underlying cold tolerance in *K. obovata*.

## 4. Materials and Methods

### 4.1. Experimental Materials

The experimental materials were collected from the research base of the Zhoushan Fisheries Research Institute, Zhejiang Province. Using the nursery seedling transplantation method, 6000 *K. obovata* seedlings were randomly divided into groups A and B and transplanted into circular plastic pots filled with sediment. Seedlings were cultivated in a nursery at a density of 12 plants/m^2^, with tidal conditions simulated. Seawater was applied twice daily, and the seedlings underwent one natural overwintering. Survival was recorded daily. The plastic pots used had an upper diameter of 25.0 cm, height of 13.0 cm, and bottom diameter of 20.0 cm, with a small circular hole at the base to prevent water accumulation.

Based on previous observations, *K. obovata* seedlings during the overwintering period were broadly classified into two groups. The first group exhibited greenish leaves and remained healthy after winter, surviving throughout the observation period; this group was defined as the cold-tolerant group. The second group showed yellowing leaves without leaf drop or withering initially, but failed to recover from overwintering damage and eventually withered; this group was defined as the cold-sensitive group. During the overwintering period, based on previous observations and the phenotypic characteristics and overall condition of the seedlings, the 3000 *K. obovata* seedlings in Group A were further classified into a cold-tolerant subgroup (A1) and -sensitive subgroup (A2), while Group B, serving as a natural population without classification, was retained as a control for Group A (Figure 12). Survival and mortality rates were calculated and the comparative results between Groups A and B are summarized in Table 1.

### 4.2. Experimental Methods

In this experiment, three ten-leaf-stage seedlings were randomly selected from each of the cold-tolerant and -sensitive groups. Surface soil and dust were gently washed off with deionized water, and the leaves were dried using paper towels. The second pair of leaves from ten-leaf-stage seedlings (consisting of five leaf pairs) was selected as the experimental material. The samples were carefully placed on a laboratory bench, photographed for comparison, and retained for records. (Figure 13). Leaf sampling was performed using scissors and forceps sterilized with 75% ethanol. Each sample was placed into pre-labeled, pre-cooled RNase-free cryotubes, immediately frozen in liquid nitrogen for 30 min, and subsequently stored at −80 °C. After collecting samples from three cold-tolerant and three cold-sensitive plants, they were designated as AL1, AL2, AL3, and BL1, BL2, BL3, respectively, serving as three biological replicates for each group.

After confirming through subsequent observations that the selected seedlings met expected phenotypic criteria, three replicates from each group were submitted for transcriptome sequencing.

Subsequently, RNA extraction and transcriptome sequencing were outsourced to Shanghai Meiji Biomedical Technology Co., Ltd. (Shanghai, China). Enzyme activities of superoxide dismutase (SOD), catalase (CAT), and peroxidase (POD) and malondialdehyde (MDA) content were measured for the cold-tolerant and -sensitive groups using commercial assay kits (Nanjing Jiancheng Bioengineering Institute (Nanjing, China)) according to the manufacturer’s protocols. All measurements were performed using fresh leaf tissue, homogenized in normal saline (NaCl 0.9%), and strictly following the instructions provided with the assay kits. Enzyme activities were expressed as units per gram of fresh tissue (U·g^−1^ FW). Chlorophyll a, chlorophyll b, and total chlorophyll contents were determined using a chlorophyll content assay kit (Suzhou Grace Biotech Co., Ltd. (Suzhou, China)).

To validate the reliability of the transcriptome sequencing results, seven DEGs were selected for quantitative real-time PCR (qRT-PCR) verification. Primers were designed for the selected genes using Primer Premier 5 software (Premier Biosoft, Palo Alto, CA, USA), with *KoACT2* [38] serving as the internal reference gene. Primer sequences are listed in Table 7. The qRT-PCR reactions were performed using the TB Green^®^ Premix Ex Taq™ II FAST qPCR Kit (TAKARA Bio Inc. (Nara, Japan)) on a CFX Duet Real-Time PCR System (Bio-Rad Laboratories, Inc. (Hercules, CA, USA)). Each 10 μL reaction contained 5 μL TB Green Premix Ex Taq II, 0.4 μL of each forward and reverse primer, 3.95 μL RNase-free water, and 0.25 μL cDNA template. The thermal cycling program consisted of an initial denaturation at 95 °C for 30 s, followed by 39 cycles of 95 °C for 5 s and 60 °C for 30 s for annealing and extension. The relative expression levels of DEGs were calculated using the 2^−ΔΔCT^ method. Statistical analyses, including analysis of variance and Pearson correlation analysis, were performed using IBM SPSS Statistics 26 (IBM Corp., Armonk, NY, USA), and graphs were generated with GraphPad Prism 8 (GraphPad Software, San Diego, CA, USA).

## 5. Conclusions

This study demonstrates that *K. obovata* exhibits multilayered physiological and molecular response mechanisms under low-temperature stress. There was significantly higher CAT activity in the cold-tolerant than the cold-sensitive group, enhancing H_2_O_2_ scavenging efficiency, whereas the cold-sensitive group exhibited higher SOD activity but insufficient downstream H_2_O_2_ clearance, resulting in lower overall antioxidant efficiency. The DEGs were primarily associated with maintenance of cellular structure, metabolic activities, redox regulation, and ion homeostasis. Significant enrichment was observed in xyloglucan metabolism, extracellular region functions, and iron ion binding. Additionally, phenylpropanoid biosynthesis, amino sugar metabolism, and plant hormone signal transduction pathways were involved in the low-temperature response. In the cold-tolerant population, TFs such as WRKY, NAC, MYB, and ERF exhibited significant differential expression, potentially activating downstream antioxidant and osmotic adjustment responses, thereby enhancing cold tolerance. The SNP and AS analyses indicated that gene mutations and splicing pattern diversity may enhance protein function and low-temperature adaptability. Overall, *K. obovata* enhances cold tolerance under low-temperature stress through multilayered mechanisms, including cellular structure protection, antioxidant regulation, maintenance of ion homeostasis, and regulation by TFs and AS. This study provides a theoretical basis for elucidating the molecular foundations of low-temperature adaptation in *K. obovata*.

In addition, the phenotypic differences observed in this study, such as leaf coloration, may provide a reference for early selection of cold-tolerant plants. The cold related differential gene markers or exploration of transcription factors obtained in this study could provide theoretical basis for selecting parental plants from the source population and further screening of cold-tolerant *K. obovata* varieties in introduced regions.

## Figures and Tables

**Figure 1 ijms-27-00429-f001:**
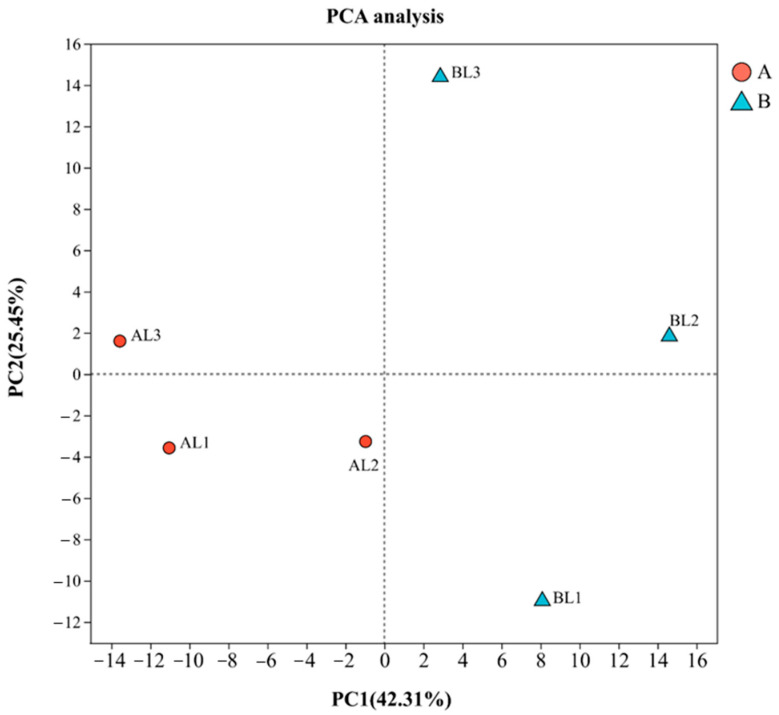
PCA of samples. PCA was performed on normalized expression data to evaluate the overall similarity and differences among samples from groups A and B. Each point represents a biological replicate (*n* = 3).

**Figure 2 ijms-27-00429-f002:**
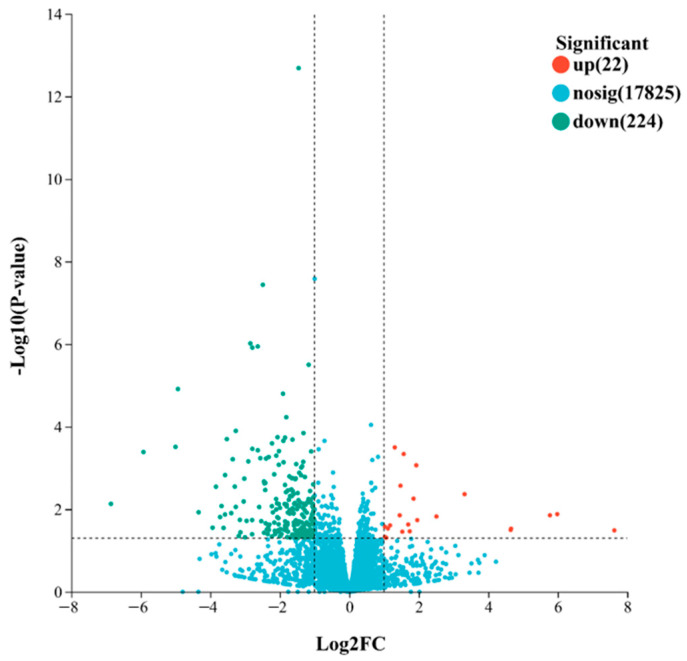
Volcano plot of DEGs between groups A and B. DEGs were defined as genes with |log_2_ fold change| ≥ 1 and adjusted *p*-value < 0.05. Dashed lines indicate thresholds for statistical significance and fold change: the horizontal dashed line corresponds to Padjust = 0.05, and the vertical dashed lines correspond to log2 fold change = ±1. Upregulated and downregulated genes are indicated in red and cyan, respectively.

**Figure 3 ijms-27-00429-f003:**
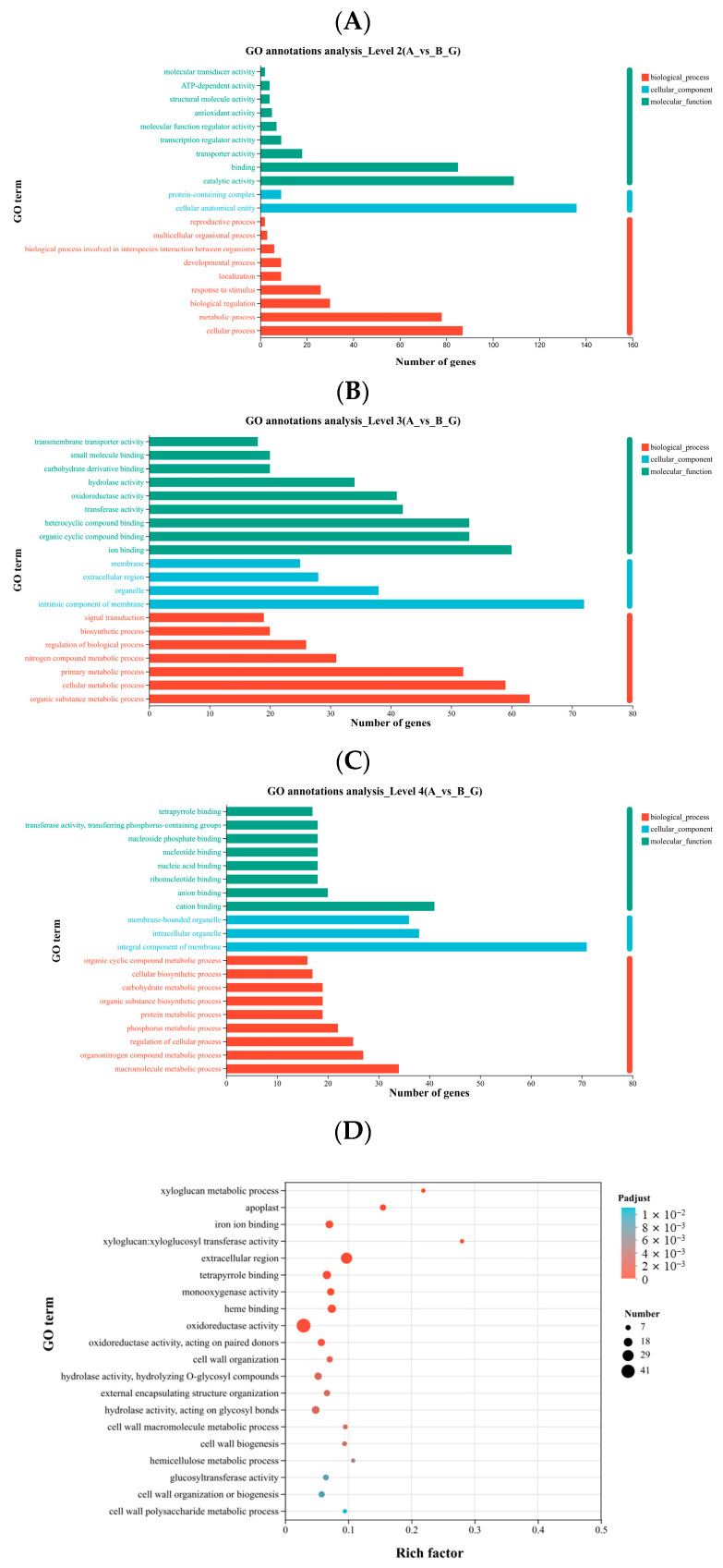
GO term annotation of DEGs between groups A and B at levels (**A**) 2, (**B**) 3 and (**C**) 4; (**D**) GO enrichment analysis of DEGs. Panels (**A**–**C**) show GO annotation: the y-axis represents GO terms, and the x-axis indicates the number of genes mapped to each term. Colors distinguish different gene sets. Panel (**D**) shows GO enrichment results. GO terms with an adjusted Padjust < 0.05 are considered significantly enriched. Bubble size represents the number of genes in each term, and color indicates the enrichment significance.

**Figure 4 ijms-27-00429-f004:**
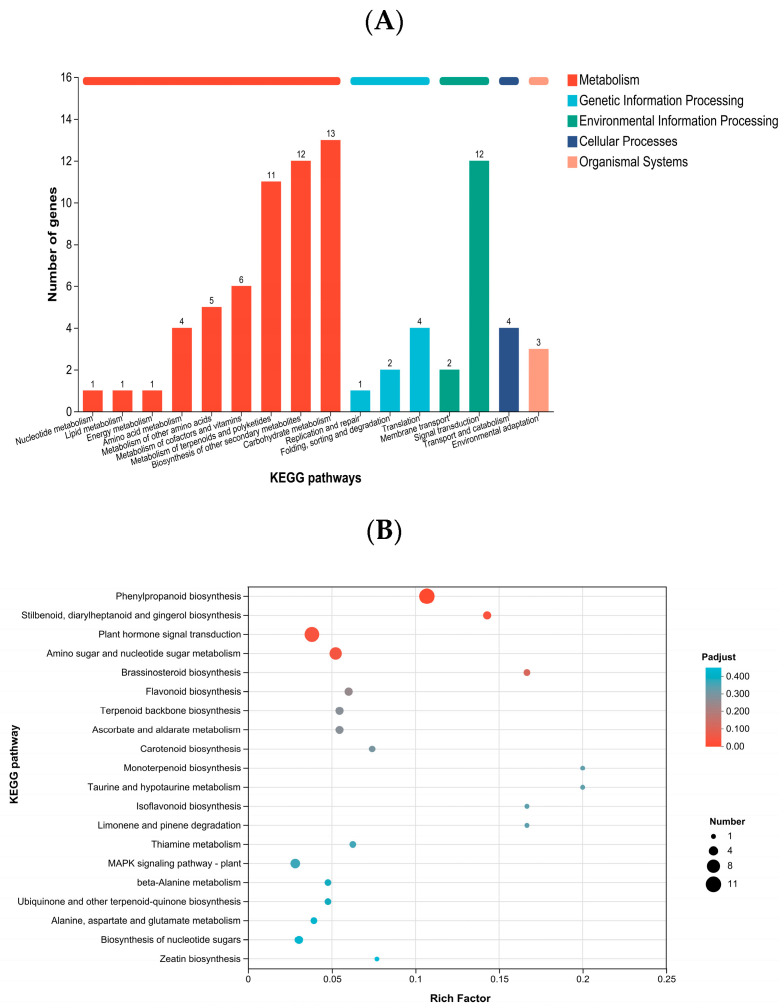
(**A**) KEGG annotation of DEGs and (**B**) KEGG enrichment analysis of DEGs between groups A and B. Panel (**A**) shows KEGG pathway annotation, with the x-axis representing pathway names and the y-axis indicating the number of genes mapped to each pathway. Panel (**B**) shows KEGG enrichment results, with pathways having an adjusted Padjust < 0.05 considered significantly enriched. Bubble size represents the number of genes in each pathway, and color indicates enrichment significance.

**Figure 5 ijms-27-00429-f005:**
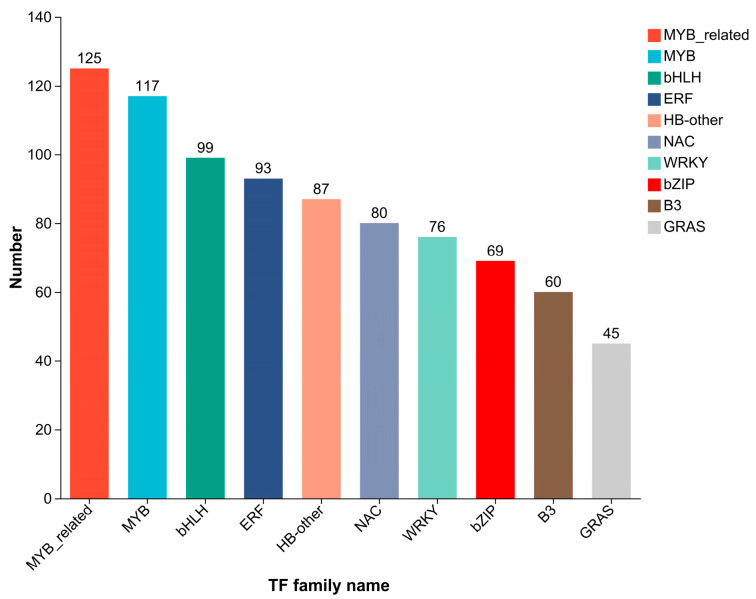
Distribution of transcription factor families in the samples. The y-axis represents the number of genes, and the x-axis indicates different TF families. Different colors are used to distinguish different TF families.

**Figure 6 ijms-27-00429-f006:**
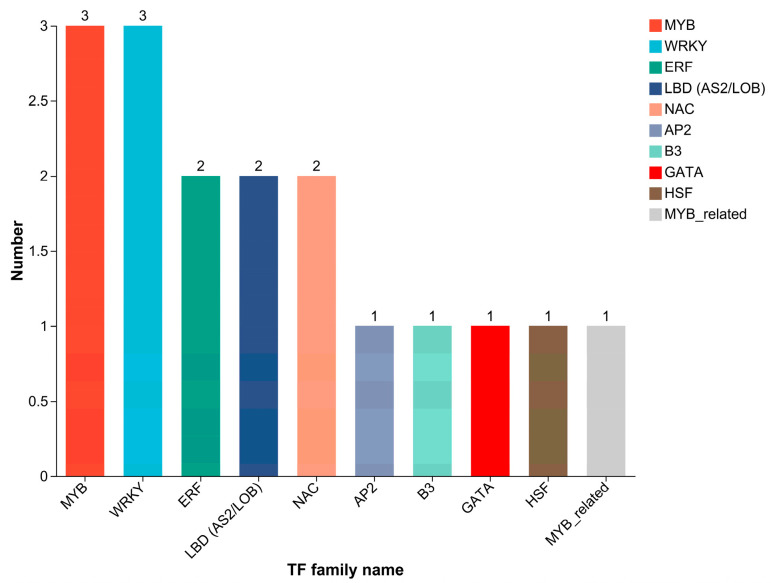
Distribution of transcription factor families among DEGs between groups A and B. The y-axis represents the number of genes, and the x-axis indicates different TF families. Different colors are used to distinguish different TF families.

**Figure 7 ijms-27-00429-f007:**
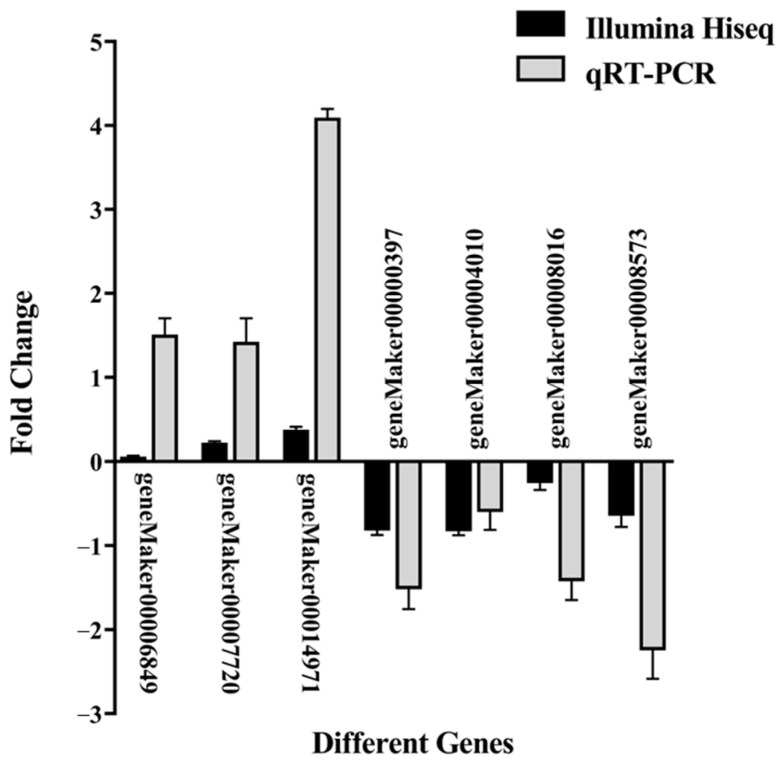
qRT-PCR validation of relative expression of selected DEGs between groups A and B. The x-axis represents different genes, and the y-axis shows log2 fold change, allowing negative values to indicate downregulation. For each gene, two bars are shown: black one for RNA-seq and gray one for qRT-PCR. Bar heights represent the mean of three biological replicates, and error bars indicate ± standard deviation (SD).

**Figure 8 ijms-27-00429-f008:**
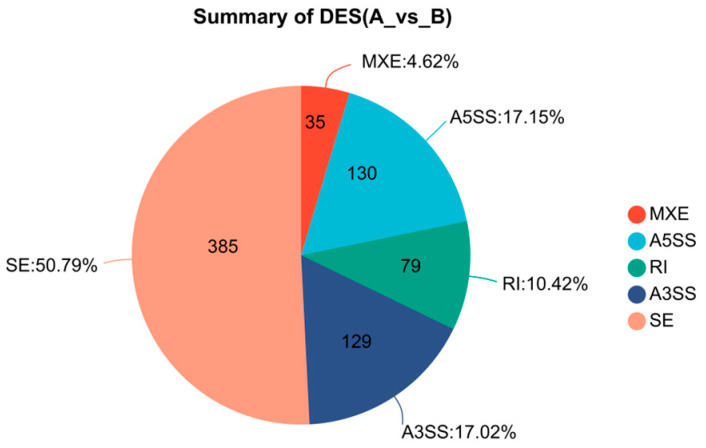
Distribution of AS events between groups A and B. The pie chart shows the proportions of five AS types: SE, MXE, A5SS, RI, and A3SS.

**Figure 9 ijms-27-00429-f009:**
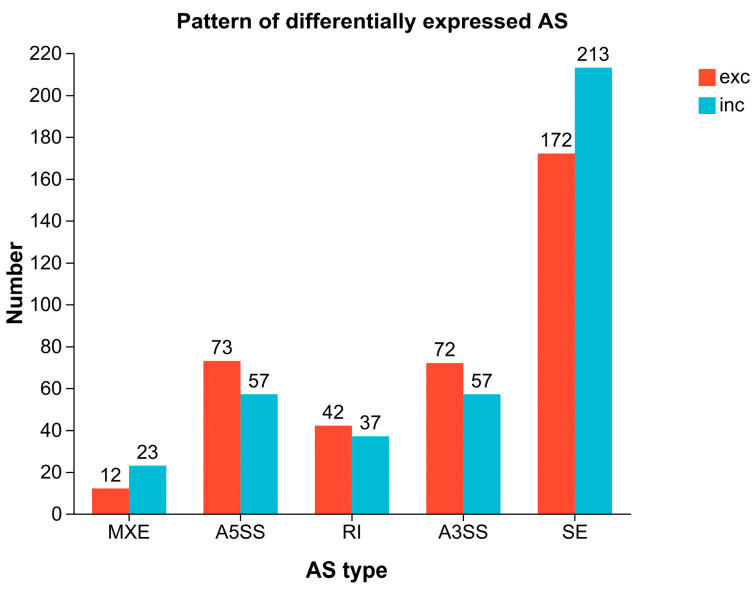
Differential AS patterns between cold-tolerant and -sensitive groups. Comparison of alternative splicing events between groups A and B. The bar chart shows the number of five AS event types (SE, MXE, A5SS, RI, and A3SS) in each group. For each AS type, two bars are shown: red one for exon exclusion (exc) and blue one for inclusion (inc).

**Figure 10 ijms-27-00429-f010:**
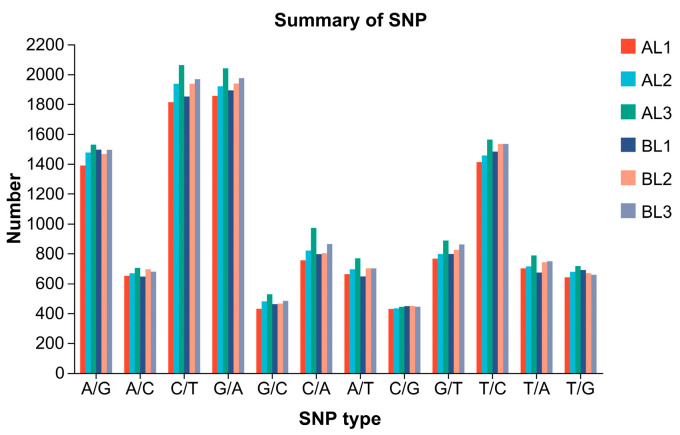
Distribution of single nucleotide polymorphism types in *K. obovata* samples from groups A and B. The x-axis represents SNP types and the y-axis represents the number of SNPs. SNPs are categorized as transitions (A↔G, C↔T) or transversions (A↔C, A↔T, G↔C, G↔T).

**Figure 11 ijms-27-00429-f011:**
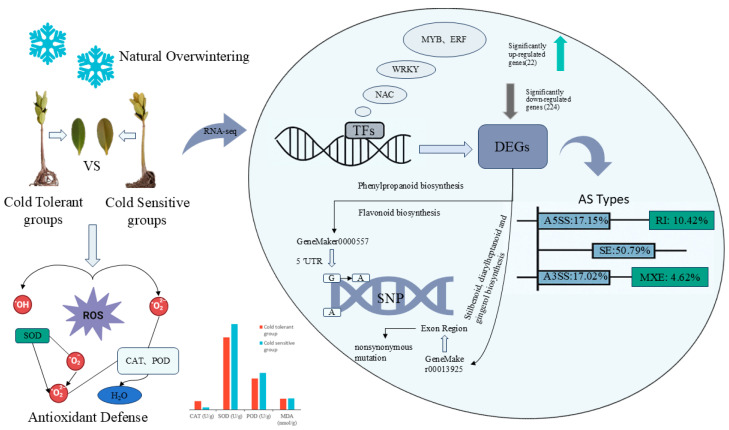
Proposed antioxidant and transcriptional regulatory mechanisms of *K. obovata* under natural overwintering conditions. The model integrates all data from this study, including sample information, measured enzyme activities and antioxidant defense indicators, and transcriptomic data such as TFs, DEGs, AS, and SNPs. Specific data sources are detailed in the Results and Discussion sections.

**Figure 12 ijms-27-00429-f012:**
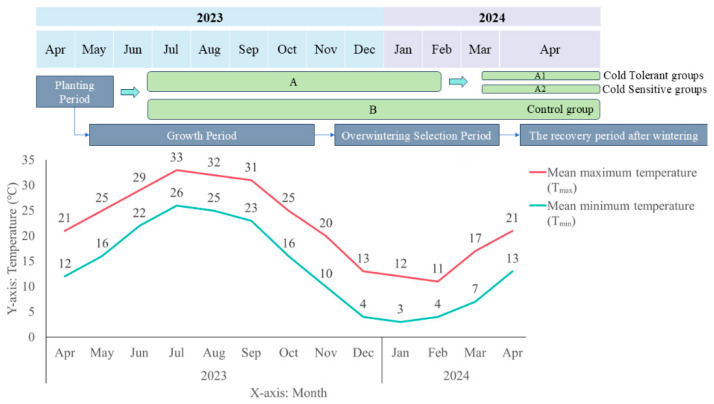
Schematic diagram of *K.obovata* planting period and key time points. The diagram indicates sample groups, sampling time points, and the overall temperature changes during the overwintering period. Temperature data were obtained from publicly available meteorological records (https://www.tianqi24.com/zhoushan/history2023.html (accessed on 4 December 2025)).

**Figure 13 ijms-27-00429-f013:**
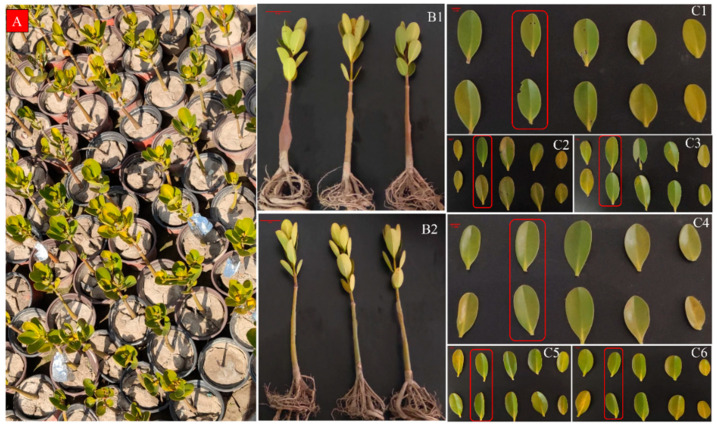
Sampling schematic of cold-tolerant and -sensitive *K. obovata* plants. (**A**) Distribution and marking of seedlings. (**B1**) Cold-tolerant and (**B2**) -sensitive *K. obovata* individuals, each group comprising three biological replicates. (**C1**–**C3**) Leaves collected from seedlings shown in (**B1**), and (**C4**–**C6**) leaves collected from seedlings shown in (**B2**). For all samples, half of the second pair of leaves was used for RNA-seq and the other half for physiological measurements.

**Table 1 ijms-27-00429-t001:** Survival rate of *K obovata* during natural overwintering.

Group	Number Planted (Plants)	Number Survived (Plants)	Survival Rate (%)	Number Dead (Plants)	Mortality Rate (%)
A1 (cold-tolerant)	1500	1339	89.3	161	10.7
A2 (cold-sensitive)	1500	86	5.72	1414	94.28
B (control)	3000	1351	45.03	1649	54.97

**Table 2 ijms-27-00429-t002:** Enzyme activities in different experimental groups.

Detection Type	Cold-Tolerant Group	Cold-Sensitive Group
Chlorophyll a (mg/g)	0.28 ± 0.05 ^a^	0.32 ± 0.02 ^a^
Chlorophyll b (mg/g)	0.68 ± 0.09 ^a^	0.70 ± 0.03 ^a^
Total chlorophyll content (mg/g)	0.96 ± 0.06 ^a^	1.01 ± 0.01 ^a^
CAT (U/g)	14.72 ± 0.19 ^a^	4.01 ± 0.89 ^b^
SOD (U/g)	1213.76 ± 66.24 ^a^	1435.10 ± 23.89 ^b^
POD (U/g)	52.63 ± 6.11 ^a^	61.89 ± 0.97 ^a^
MDA (nmol/g)	18.40 ± 0.83 ^a^	19.37 ± 0.81 ^a^

Note: different superscript letters in the same row indicate significant differences (*p* < 0.05).

**Table 3 ijms-27-00429-t003:** Quality assessment of transcriptome sequencing data.

Sample	Clean Reads	Clean Bases	Error Rate (%)	Q20 (%)	Q30 (%)	GC Content (%)
AL1	42,780,366	6,437,354,464	0.0118	98.90	96.45	45.93
AL2	46,537,646	6,992,527,814	0.0117	98.96	96.63	45.71
AL3	49,719,668	7,469,910,026	0.0116	99.00	96.77	45.84
BL1	44,857,924	6,742,092,163	0.0116	98.98	96.73	45.45
BL2	50,516,336	7,585,021,404	0.0115	99.02	96.86	45.60
BL3	45,178,010	6,789,571,579	0.0116	98.99	96.74	45.64

**Table 4 ijms-27-00429-t004:** Mapping statistics of transcriptome sequencing reads to the reference genome.

Sample	Clean Reads	Total Mapped (%)	Multiple Mapped (%)	Uniquely Mapped (%)
AL1	42,780,366	97.95	3.39	94.56
AL2	46,537,646	97.89	3.67	94.22
AL3	49,719,668	97.88	3.73	94.15
BL1	44,857,924	97.79	3.96	93.83
BL2	50,516,336	97.93	3.39	94.54
BL3	45,178,010	98.12	3.16	94.97

**Table 5 ijms-27-00429-t005:** Top five KEGG pathways with significant enrichment in DEGs of *K. obovata*.

KEGG Pathway	Pathway ID	Number of DEGs	*p*-Value	Padjust	First Category	Second Category
Phenylpropanoid biosynthesis	map00940	11	5.98 × 10^−8^	3.23 × 10^−6^	Metabolism	Biosynthesis of other secondary metabolites
Amino sugar and nucleotide sugar metabolism	map00520	7	1.50 × 10^−3^	0.0404	Metabolism	Carbohydrate metabolism
Plant hormone signal transduction	map04075	10	1.76 × 10^−3^	0.0317	Environmental Information Processing	Signal transduction
Stilbenoid, diarylheptanoid and gingerol biosynthesis	map00945	3	2.21 × 10^−3^	0.0299	Metabolism	Biosynthesis of other secondary metabolites
Brassinosteroid biosynthesis	map00905	2	9.62 × 10^−3^	0.104	Metabolism	Metabolism of terpenoids and polyketides

**Table 6 ijms-27-00429-t006:** Summary of single nucleotide polymorphism types across experimental samples.

Type		AL1	AL2	AL3	BL1	BL2	BL3
Transition (Ti)	A/G	1388	1474	1529	1493	1464	1492
	C/T	1814	1936	2060	1852	1936	1967
	G/A	1856	1919	2038	1892	1937	1974
	T/C	1411	1454	1563	1480	1534	1534
Total		6469	6783	7189	6717	6871	6967
Transversion (Tv)	A/C	650	667	703	645	694	678
	G/C	429	479	525	460	464	482
	C/A	754	818	971	794	800	862
	A/T	661	694	766	646	700	700
	C/G	428	432	442	447	448	443
	G/T	764	795	885	795	823	859
	T/A	700	713	785	672	741	748
	T/G	640	677	715	689	668	657
Total		5026	5275	5793	5148	5338	5429
Ti/Tv		1.29	1.29	1.24	1.30	1.29	1.28
Total		11,495	12,058	12,982	11,865	12,209	12,396

**Table 7 ijms-27-00429-t007:** Primers sequences used in the qRT-PCR.

Gene ID	Forward Primer (5′→3′)	Reverse Primer (5′→3′)
MYB-related-14971	GCAGGTTCGGAGCAGCAATTATC	ACATCTGATTTAGTGCGTTCGTTGG
MYB-7720	TTCAGACCTTCTACCGACAACCATC	GAGTGGCTGCTGAGTTTGTTTGG
LBD (AS2/LOB)-6849	GCTGTGGTGGATGGCGAGAG	TGTTGGCATCACGGACTCCTTG
LBD (AS2/LOB)-8016	TTTCCTCCTTGAACCCTCACATCTC	CCCAGGTCACACTAGCACACTATTC
GATA-397	CGACCACATTGACGACCTCCTC	ACTCGGACTCAGCAGACCAAATG
ERF-4010	CCATCGGGACCACCAACTAAAGG	AGGCGACCACGTATCCAATGC
ERF-8573	TGGATCGAAGACATGGCAAGAGG	GTTGAGATGGAGCGGAGTAGATGG
KoACT2 (internal reference)	ACCGAGGCTCCTCTTAATCC	AGCTGGCACATTGAAGGTCT

## Data Availability

The data presented in this study are available on request from the corresponding author.

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
