# Peer review of "Research on Cold Resistance of Kandelia obovata Transplanted to Zhoushan Area at the mRNA Level"

_ijms, 2025, doi:10.3390/ijms27010429_

Round 1

Reviewer 1 Report

Comments and Suggestions for Authors

Several aspects require clearer description and/or stronger framing to ensure that the conclusions are well-supported and reproducible.

Major points

1. The Methods state that three seedlings per group were sampled and used as three biological replicates (AL1–AL3 and BL1–BL3). Please explicitly define:

- What constitutes one biological replicate (one plant? pooled leaves from multiple plants? multiple leaf punches combined?) 

- Whether enzyme activities and chlorophyll were measured on the same individuals used for transcriptomics.

- If pooling was performed (e.g., several plants per replicate), state the exact number. If not pooled, acknowledge the limited n and its implications.

2. Table 2 reports CAT/SOD/POD in U/g (P. 4), but it is unclear whether this is per gram fresh weight, dry weight, or protein. Please specify the basis and add the missing methodological details (e.g., protein quantification if activities are expressed as U·mg⁻¹ protein). This is essential for comparability across studies.

3. Plants are classified as “cold-tolerant” vs “cold-sensitive” based on overwintering outcomes/phenotypes. This is reasonable, but it also creates interpretive constraints. The transcriptome comparison may capture not only cold-response differences but also downstream consequences of decline/senescence in failing plants (the manuscript itself notes PCA separation is imperfect). Please strengthen the Discussion by explicitly stating these limitations and avoiding causal wording unless supported by functional validation.

Minor points

  1. Near the end of the Introduction, add 1–2 sentences explicitly stating what is new here (e.g., natural overwintering survival phenotype + integrated antioxidant enzyme activity + transcriptome + exploratory SNP/AS in introduced high-latitude plants).
  2. At the end of Conclusions, briefly state how these results could be used (e.g., guidance for screening cold-tolerant planting material; candidate TF/DEG markers for breeding; management of introduction sites). The Conclusions already summarize mechanisms, but the applied “why it matters” could be clearer.
  3. Ensure consistent species naming (Kandelia obovata, not “obovate”) - P. 1.

Author Response

Dear Reviewer,

Thank you very much for taking the time to review our manuscript and for providing your valuable comments. We have revised the manuscript according to your suggestions and have highlighted the changes. Please refer to the responses below and the revised manuscript for details.

Comment 1:

The Methods state that three seedlings per group were sampled and used as three biological replicates (AL1–AL3 and BL1–BL3). Please explicitly define:

What constitutes one biological replicate (one plant? pooled leaves from multiple plants? multiple leaf punches combined?).

Whether enzyme activities and chlorophyll were measured on the same individuals used for transcriptomics.

If pooling was performed (e.g., several plants per replicate), state the exact number. If not pooled, acknowledge the limited n and its implications.

Response 1:

We sincerely thank the reviewer for this important comment.

For “What constitutes one biological replicate (one plant? pooled leaves from multiple plants? multiple leaf punches combined?)

The definition of biological replicates has been further clarified in the Methods section of the revised manuscript (Section 4.2, p.17, Lines 3–11). Specifically, each sample was derived from a single ten -leaf-stage (five pairs of leaves) Kandelia obovata seedling, and each treatment group consisted of three independent seedlings (AL1–AL3 and BL1–BL3) as biological replicates.

For “Whether enzyme activities and chlorophyll were measured on the same individuals used for transcriptomics.

Moreover, the material used for physiological measurements and transcriptome sequencing was taken from the same portion of the second pair of leaves of each seedling, ensuring consistency across analyses.

For “If pooling was performed (e.g., several plants per replicate), state the exact number. If not pooled, acknowledge the limited n and its implications.

We also acknowledge that the experimental design limited the sample size (n = 3), and the potential impact of this limitation has been discussed in the Discussion section (Section 3.1, p.14, 2nd paragraph Lines 8–10).

Comment 2:

Table 2 reports CAT/SOD/POD in U/g (P. 4), but it is unclear whether this is per gram fresh weight, dry weight, or protein. Please specify the basis and add the missing methodological details (e.g., protein quantification if activities are expressed as U·mg⁻¹ protein). This is essential for comparability across studies.

Response 2:

We sincerely thank the reviewer for pointing out this issue. The calculation basis for enzyme activities has been clarified in the Methods section of the revised manuscript (Section 4.2, p. 18, 1st paragraph, Lines 6–8). CAT, SOD and POD are expressed per gram of fresh tissue (U·g⁻¹ FW). All measurements were performed using fresh leaf tissue, homogenized in physiological saline, and strictly following the instructions provided with the assay kits. These details have been clearly added to the manuscript to improve the clarity of the experimental methods and to ensure comparability with other studies.

Comment 3:

Plants are classified as “cold-tolerant” vs “cold-sensitive” based on overwintering outcomes/phenotypes. This is reasonable, but it also creates interpretive constraints. The transcriptome comparison may capture not only cold-response differences but also downstream consequences of decline/senescence in failing plants (the manuscript itself notes PCA separation is imperfect). Please strengthen the Discussion by explicitly stating these limitations and avoiding causal wording unless supported by functional validation.

Response 3:

We sincerely thank the reviewer for this valuable comment and fully agree that classifying plants based on natural overwintering phenotypes may impose certain limitations on result interpretation. To minimize potential confounding effects caused by leaf senescence or severe tissue damage, transcriptome sampling was performed at the early recovery stage after winter, prior to obvious tissue collapse or widespread leaf senescence. During sampling, visibly wilted leaves were avoided as controls. Nevertheless, some transcriptomic differences may still reflect early downstream processes associated with subsequent plant decline. Accordingly, we have explicitly discussed these limitations in the Discussion section (Section 3.1, p. 13, 2nd paragraph, Lines 4–10) and have refrained from making causal inferences about gene functions or regulatory relationships in the absence of functional validation.

Comment 4:

Near the end of the Introduction, add 1–2 sentences explicitly stating what is new here (e.g., natural overwintering survival phenotype + integrated antioxidant enzyme activity + transcriptome + exploratory SNP/AS in introduced high-latitude plants).

Response 4:

We sincerely thank the reviewer for this valuable suggestion and agree that it is important to more clearly highlight the novelty of our study. Accordingly, we have added a brief statement at the end of the Introduction (Section 1, p. 3, Lines 13-19) emphasizing the innovative aspects of this work. Specifically, our study combines physiological measurements and transcriptome analysis of K. obovata under natural overwintering conditions. By comparing antioxidant enzyme activities and gene expression differences at the mRNA level between cold-tolerant and -sensitive plants, we provide preliminary insights into the mechanisms of cold resistance. In addition, by exploring potential SNPs and AS events, this study offers a theoretical basis for introducing K. obovata to higher-latitude regions and for future breeding programs.

Comment 5:

At the end of Conclusions, briefly state how these results could be used (e.g., guidance for screening cold-tolerant planting material; candidate TF/DEG markers for breeding; management of introduction sites). The Conclusions already summarize mechanisms, but the applied “why it matters” could be clearer.

Response 5:

We sincerely thank the reviewer for this valuable suggestion. We have added a statement at the end of the Conclusion section (Section 5, p.18, 2nd paragraph) to clarify the potential applications of our findings. For example, the phenotypic differences observed in this study, such as leaf coloration, may provide a reference for early selection of cold-tolerant plants. Furthermore, the identification of cold-related differential gene markers or the exploration of transcription factors could offer a theoretical basis for selecting parental plants from source populations and for further screening of cold-tolerant K. obovata varieties in introduced regions.

Comment 6:

Ensure consistent species naming (Kandelia obovata, not “obovate”) - P. 1.

Response 6:

Thank you for pointing out the issue regarding the species name. The incorrect statement on the first page has been corrected, and a thorough check of the entire manuscript has been conducted to ensure that the species name Kandelia obovata is used correctly and consistently throughout the text.

Reviewer 2 Report

Comments and Suggestions for Authors

hi, thank you for your effort in this study. I have really enjoyed the manuscript. however i have some suggestion-

  1. Please provide a more detailed description of the qRT-PCR protocol, including the number of amplification cycles performed.

  2. The figure size should be increased to improve readability and visibility of the data.

  3. The graphical presentation of the results should be enhanced to ensure greater clarity and visual quality.
    Thanks.

Author Response

Dear Reviewer,

Thank you very much for taking the time to review our manuscript and for providing your valuable comments. We have revised the manuscript according to your suggestions and have highlighted the changes in the text. Please refer to the responses below and the revised manuscript for details.

Comment 1:

Please provide a more detailed description of the qRT-PCR protocol, including the number of amplification cycles performed.

Response 1:

We sincerely thank the reviewer for this valuable suggestion. We have added a more detailed description of the qRT-PCR procedure in the Materials and Methods section (Section 4.2, p.18, 2nd paragraph, lines 4–10), specifying the amplification conditions, including the total number of PCR cycles and the specific parameters for denaturation, annealing, and extension.

Comment 2:

The figure size should be increased to improve readability and visibility of the data.

Response 2:

We sincerely thank the reviewer for highlighting this issue. In the revised manuscript, we have enlarged the overall size of the relevant figures and adjusted the font sizes of the axis labels and legends accordingly. These modifications improve the clarity and readability of the figures, ensuring that the data are accurately presented in both electronic and printed versions.

Comment 3:

The graphical presentation of the results should be enhanced to ensure greater clarity and visual quality. Thanks.

Response 3:

We sincerely thank the reviewer for pointing out this issue. Following your suggestion, we have optimized the presentation of the result figures in the revised manuscript, including improving figure layouts, increasing resolution, and enlarging the font size of labels and text within the figures. These modifications allow the research results to be presented more clearly and intuitively.

Reviewer 3 Report

Comments and Suggestions for Authors

The authors investigated the physiological and molecular mechanisms underlying cold tolerance in the mangrove species Kandelia obovate by measuring the content of photosynthetic pigments and the activities of antioxidant enzymes, as well as analyses of differentially expressed genes, transcription factor families, single nucleotide polymorphisms, and alternative splicing events. Post-winter individuals from the same batch of K. obovata seedlings were divided into two groups, based on phenotypic characteristics and their general condition – cold-tolerant (survived winter) and cold-sensitive (died after winter). New data on the molecular mechanisms underlying the cold tolerance of K. obovate are presented. The results provide a scientific basis for the introduction of K. obovata to northern Zhejiang and the cultivation of cold-tolerant varieties.

I have the following comments and questions:

The results showed no significant differences between the two K. obovata groups in terms of chlorophyll content and lipid peroxidation levels. Catalase activity was higher in the cold-tolerant group, while superoxide dismutase activity was higher in the cold-sensitive group. The lack of differences in physiological parameters is explained by the sampling during the early stage of wintering. When were the measurements performed? What exactly does mean “during the early phase of low-temperature” or “during the early overwintering stage”? Why didn’t you measure changes in physiological parameters at a later stage as well, but only during the early stage of overwintering?

What is the purpose of having a group B, in addition to group A, which also includes cold-resistant and sensitive plants? Only the survival rate of plants in this group is presented in Table 1.

In the introduction it is written that “Kandelia obovata, of the family Rhizophoraceae, is the most widely distributed and cold-tolerant mangrove species in China…” and 2 sentences below “Temperature can influence the distribution of temperature-sensitive plants such as K. obovate” – I guess what you mean, but you should write it clearly.

The text in Figure 3 is unreadable. Is it possible to increase the letter size? Please, check the size of letters in all figures.

Author Response

Dear Reviewer,

Thank you very much for taking the time to review our manuscript and for providing your valuable comments. We have revised the manuscript according to your suggestions and have highlighted the changes in the text. Please refer to the responses below and the revised manuscript for details.

Comment 1:

When were the measurements performed? What exactly does mean “during the early phase of low-temperature” or “during the early overwintering stage”? Why didn’t you measure changes in physiological parameters at a later stage as well, but only during the early stage of overwintering?

Response 1:

We sincerely thank the reviewer for this valuable suggestion. All leaf samples were collected in February 2024 (early recovery stage after winter) and immediately subjected to physiological measurements in the laboratory following the standardized procedure provided with the assay kits.

The term “early post-winter stage” in the manuscript originally referred to the initial antioxidant enzyme defense responses of Kandelia obovata under natural low-temperature stress conditions in the Zhoushan area, and this has been clarified in the revised manuscript (Section 3.1, p. 13, 4th paragraph, line 1). Under low-temperature stress, K. obovata ceases growth, and the physiological performance of the leaves reflects the cold response. Cold-sensitive plants may experience mortality after winter, resulting in large physiological variability at that stage. Therefore, in this study, enzyme activities were measured only during the early recovery stage after winter (February 2024).

Comment 2:

What is the purpose of having a group B, in addition to group A, which also includes cold-resistant and sensitive plants? Only the survival rate of plants in this group is presented in Table 1.

Response 2:

We sincerely thank the reviewer for raising this question. The purpose of Group B was to provide a reference for Group A, in order to evaluate the feasibility of classifying K. obovata seedlings into cold-tolerant and cold-sensitive groups based on natural overwintering phenotypes, and to assess whether early selection of seedlings would affect overall survival rates. Group A represents the experimental group divided according to predefined criteria, while Group B reflects the natural population and was primarily used for statistical comparison of overwintering survival rates. (Section 4.1, p. 17, lines 4–9).

Comment 3:

In the introduction it is written that “Kandelia obovata, of the family Rhizophoraceae, is the most widely distributed and cold-tolerant mangrove species in China…” and 2 sentences below “Temperature can influence the distribution of temperature-sensitive plants such as K. obovate” – I guess what you mean, but you should write it clearly.

Response 3:

We sincerely thank the reviewer for pointing out this issue. We have revised the relevant statements to improve rigor and logical coherence. The revised text now reads: “Kandelia obovata, of the family Rhizophoraceae, is the most widely distributed and cold-tolerant mangrove species in China, occurring at the highest latitudes among mangroves. However, its distribution is still limited by low temperatures. .” (Section 1, p. 2, 2nd paragraph, lines 1-3)

Comment 4:

The text in Figure 3 is unreadable. Is it possible to increase the letter size? Please, check the size of letters in all figures.

Response 4:

We sincerely thank the reviewer for pointing out this issue. We have checked the font sizes of all figures in the manuscript and adjusted the sizes of the individual panels in Figure 3 to ensure that the text is clear and legible. Additionally, the font sizes of all other figures have been standardized to improve overall readability.